# Association between plaque vulnerability and neutrophil extracellular traps (NETs) levels: The Plaque At RISK study

Judith J. de Vries[1,◉], Anouchska S. A. Autar[1,2,◉], Dianne H. K. van Dam-Nolen[3], Samantha J. Donkel[1], Mohamed Kassem[4], Anja G. van der Kolk[5,6], Twan J. van Velzen[7], M. Eline Kooi[4], Jeroen Hendrikse[6], Paul J. Nederkoorn[7], Daniel Bos[3,8], Aad van der Lugt[3], Moniek P. M. de Maat[1], Heleen M. M. van Beusekom[2]*

1 Department of Hematology, Erasmus MC Rotterdam, University Medical Center Rotterdam, Rotterdam, The Netherlands, 2 Department of Experimental Cardiology, Erasmus MC Rotterdam, University Medical Center Rotterdam, Rotterdam, The Netherlands, 3 Department of Radiology and Nuclear Medicine, Erasmus MC Rotterdam, University Medical Center Rotterdam, Rotterdam, The Netherlands, 4 Department of Radiology and Nuclear Medicine, CARIM School for Cardiovascular Diseases, Maastricht University Medical Centre, Maastricht, The Netherlands, 5 Department of Radiology, Netherlands Cancer Institute / Antoni van Leeuwenhoek Hospital, Amsterdam, The Netherlands, 6 Department of Radiology, University Medical Center Utrecht, Utrecht, The Netherlands, 7 Department of Neurology, Academic Medical Centre, Amsterdam, The Netherlands, 8 Department of Epidemiology, Erasmus MC, University Medical Center Rotterdam, Rotterdam, The Netherlands

◉ These authors contributed equally to this work.
* h.vanbeusekom@erasmusmc.nl

**Data Availability Statement:** The data underlying this article cannot be shared publicly due to the privacy of individuals that participated in this study.

## Abstract

Carotid atherosclerotic plaque rupture and its sequelae are among the leading causes of acute ischemic stroke. The risk of rupture and subsequent thrombosis is, among others, determined by vulnerable plaque characteristics and linked to activation of the immune system, in which neutrophil extracellular traps (NETs) potentially play a role. The aim of this study was to investigate how plaque vulnerability is associated with NETs levels. We included 182 patients from the Plaque At RISK (PARISK) study in whom carotid imaging was performed to measure plaque ulceration, fibrous cap integrity, intraplaque hemorrhage, lipid-rich necrotic core, calcifications and plaque volume. Principal component analysis generated a 'vulnerability index' comprising all plaque characteristics. Levels of the NETs marker myeloperoxidase-DNA complex were measured in patient plasma. The association between the vulnerability index and low or high NETs levels (dependent variable) was assessed by logistic regression. No significant association between the vulnerability index and NETs levels was detected in the total population (odds ratio 1.28, 95% confidence interval 0.90–1.83, $p = 0.18$). However, in the subgroup of patients naive to statins or antithrombotic medication prior to the index event, this association was statistically significant (odds ratio 2.08, 95% confidence interval 1.04–4.17, $p = 0.04$). Further analyses revealed that this positive association was mainly driven by intraplaque hemorrhage, lipid-rich necrotic core and ulceration. In conclusion, plaque vulnerability is positively associated with plasma levels of NETs, but only in patients naive to statins or antithrombotic medication prior to the index event.

The data will be shared on reasonable request to the Radiology trial office (imaging. trialbureau@erasmusmc.nl).

**Funding:** This work was supported by the Centre for Translational Molecular Medicine (www.ctmm. nl) [project PARISK (Plaque at RISK); grant 01C-202]; the Dutch Heart Foundation (www. hartstichting.nl), the Trombose stichting (www. trombosestichting.nl) [TSN2013-2]; and the European Union's Horizon 2020 – Research and Innovation Framework Programme (https://ec. europa.eu/programmes/horizon2020/) [Marie Skłodowska-Curie grant agreement No 722609]. The funders had no role in study design, data collection and analysis, decision to publish, or preparation of the manuscript.

**Competing interests:** I have read the journal's policy and the authors of this manuscript have the following competing interests: MEK reports grants outside the submitted work from NWO Aspasia, NWO Hestia, and EU Horizon 2020 ITN and has research collaborations with PieMedical Systems, Machnet BV. This does not alter our adherence to PLOS ONE policies on sharing data and materials.

# Introduction

Ischemic stroke is a major contributor to disability and mortality worldwide [1]. An important risk factor for the development of ischemic stroke and recurrent events is the presence of vulnerable atherosclerotic plaques in the carotid arteries [2]. Vulnerable plaque characteristics, which make plaques more prone to rupture and are related to ischemic events, are intraplaque hemorrhage (IPH), lipid-rich necrotic core (LRNC), plaque ulcerations, and a thin or ruptured fibrous cap (TRFC) [3, 4].

Increasing evidence suggests that inflammation plays a pivotal role in the development of plaque vulnerability and that vulnerable plaques can activate the immune system [5]. One specific inflammatory process that has been suggested to be associated with the vulnerability of plaques is the release of neutrophil extracellular traps (NETs) by activated neutrophils [6]. NETs are composed of DNA, citrullinated histones and granular proteins, such as myeloperoxidase (MPO) [7]. NETs are present in human atherosclerotic lesions and thrombi [8], and plasma levels of NETs markers are increased in patients with acute ischemic stroke as compared to healthy subjects [9].

The underlying causal mechanism relating NETs to ischemic stroke is not completely clear. NETs might be released as a result of a vulnerable atherosclerotic environment, while on the other hand NETs can also promote atherogenesis or destabilization of plaques [10]. It has also been shown that NETs play a role in thrombosis by forming a scaffold for clot formation and promoting gene expression and activation of coagulation factors, which could connect NETs and thrombosis after plaque rupture [11–13].

To further elucidate the mechanism underlying the increased NETs formation in ischemic events, we hypothesized that plaque characteristics that make atherosclerotic plaques more vulnerable are associated with increased levels of NETs. In the current study, our primary aim was to determine the association between plaque vulnerability and plasma levels of NETs in patients with a recent transient ischemic attack (TIA) or ischemic stroke and proven atherosclerotic disease in the carotid artery. Secondly, we investigated which specific vulnerable plaque characteristics are driving potential associations between plaque vulnerability and plasma levels of NETs.

# Methods

## Study population

We studied patients participating in the PARISK study (Plaque At RISK; clinical trials.gov NCT01208025). The PARISK study is a prospective multicenter cohort study using non-invasive plaque imaging to identify which patients with a mild-to-moderate carotid artery stenosis have an increased risk of recurrent stroke [14]. Patients were enrolled between September 2010 and December 2014 and within three months after experiencing neurological symptoms due to ischemia (TIA, including amaurosis fugax, or minor stroke (modified Rankin scale ≤3)). To be included in the study, stenosis of the ipsilateral carotid artery should be above 30% (based on the European Carotid Surgery Trial) and below 70% (based on the North American Symptomatic Carotid Endarterectomy Trial) [15, 16]. Exclusion criteria were a probable cardiac source of embolism, clotting disorders, standard contraindications for magnetic resonance imaging (MRI) and severe comorbidities complicating the visit to the hospital. Patients with a renal clearance of $< 30$ mL/min/1.73 m$^2$ or an MRI contrast allergy did not receive the MRI contrast agent. Patients having a renal clearance of $< 60$ mL/min/1.73 m$^2$ or a contrast allergy did not undergo the multidetector-row computed tomography angiography (MDCTA) scan. The PARISK study was subject to the Medical Research Involving Human Subjects Act

and approved by the Medical Ethics Committees of the Academic Medical Center Amsterdam, the Erasmus University Medical Center Rotterdam, the Maastricht University Medical Center and the University Medical Center Utrecht (registration number NL29116.068.09 / MEC 09-2-082). All patients provided written informed consent. In the current study, all patients from the PARISK study with carotid imaging and an available blood sample were included (n = 182). Clinical baseline data such as age, sex, type of ischemic event, medication use, medical history and cardiovascular risk factors were collected and defined as reported before [17].

## MRI and MDCTA acquisition and analysis

Standardized MDCTA and multi-sequence contrast-enhanced MRI protocols were used, as described in the study design paper [14]. All imaging studies were evaluated by trained readers blinded for clinical data and other imaging tests [18]. MDCTA images were reviewed using dedicated 3D analysis software (Syngo.via; Siemens, Erlangen, Germany) and rated on image quality [18]. Secondly, we assessed the presence of plaque ulceration in the symptomatic artery, defined as extension of contrast material of $\geq$ 1 mm into the atherosclerotic plaque on at least two orthogonal planes [19, 20]. Furthermore, the size of the ulceration was classified into three groups (1 mm, between 1 and 2 mm or > 2 mm). In addition, the degree of stenosis and calcifications in the symptomatic carotid artery were determined as described before [17]. MRI images were evaluated with dedicated vessel wall analysis software (VesselMASS, Department of Radiology, Leiden University Medical Center, the Netherlands) as described previously [18]. IPH, LRNC, TRFC and plaque volume were derived from the MRI images as described earlier [17]. Relative volumes of IPH, LRNC and calcification were defined as percentage of plaque volume [21].

## Blood sampling and measurements

Platelet-poor plasma was obtained by centrifugation of citrated blood (3.2% sodium citrate) at 2,000g for 10 minutes at room temperature, followed by centrifugation at 14,000g for 10 minutes and stored in aliquots at -80˚C until further measurements. As a marker for NETs, MPO-DNA complexes were measured using a capture enzyme-linked immunosorbent assay (ELISA) as reported previously [22] by adjusting the commercial human cell death ELISA kit (Cell Death Detection ELISA$^{PLUS}$, Cat. No. 11774425002; Roche Diagnostics, Almere, the Netherlands). The mouse anti-human MPO monoclonal antibody (clone 4A4, Cat. No. 0400–002; Bio-Rad, CA, USA) was used as capture antibody. Patient plasma was incubated with the peroxidase-labeled anti-DNA monoclonal antibody (clone MCA-33) from the Cell Death Detection ELISA$^{PLUS}$ kit, after which peroxidase substrate was added and absorption was measured at 405 nm using the Biotek Synergy HT plate reader. Values are expressed as milli-arbitrary units (mAU), based on reference samples. The reference line was prepared by incubating isolated neutrophils from 3 healthy donors for 2,5 hours with 250 ng/ml phorbol 12-myristate 13-acetate (Sigma Aldrich, Amsterdam, the Netherlands) in plasma to induce NETs formation, as described previously [23, 24].

Histone-complexed DNA fragments (histone-DNA), a marker of cell death in general and not neutrophil specific, were measured using the commercial human cell death ELISA kit, following the manufacturer's instructions (Cell Death Detection ELISA$^{PLUS}$, Cat. No. 11774425002, Roche Diagnostics, Almere, the Netherlands).

## Statistical analysis

Baseline data are presented as mean with standard deviation (SD) for normally distributed data, median with 25th-75th percentile for skewed data, or number with percentage (%) for

categorical data. MPO-DNA complex and histone-DNA levels had a strongly skewed distribution (S1 Fig), which could not be corrected by transformation; patients were therefore classified as having low or high levels to be able to use them as dependent variable in regression models. The cut-off was set at 50% of the number of observations, resulting in two groups of similar sizes with statistically significant different levels of both markers (S1 Fig).

To assess the association between plaque vulnerability and plasma levels of NETs, we first combined the different plaque characteristics into one 'vulnerability index', which we constructed using principal component analysis. Using the varimax rotation method, the following plaque characteristics were submitted to the analysis: relative IPH volume, relative LRNC volume, relative calcification volume, ulceration size, presence of TRFC, and plaque volume. The component with the highest eigenvalue was selected and for each patient a factor score was calculated using the factor loadings [25]. Next, the association between the vulnerability index (independent variable) and the plasma levels of NETs (dependent variable) was tested using logistic regression models while adjusting for age, sex and time between index event and blood sampling. The group with low levels of NETs was used as reference category. Secondary, if a significant association was found between the vulnerability index and NETs levels, we assessed which of the individual plaque characteristics were associated with plasma levels of NETs, using logistic regression. Medication use, and more specifically statins and antithrombotic medication, is known to affect plaque vulnerability and inflammation. We observed significant interactions between vulnerable plaque characteristics and prior use of this medication prior to the index event. Therefore, we also performed subgroup analyses in patients with or without medication use prior to the index event. The difference in patients characteristics between the two subgroups were tested by the independent students' t-test for normally distributed variables, Mann-Whitney U test for not-normally distributed variables or the Chi-square test for categorical variables. A $p$-value below 0.05 was considered statistically significant. All statistical analyses were performed using IBM SPSS Statistics for Windows, version 25 (IBM Corp., Armonk, N.Y., USA).

## Results

Baseline clinical characteristics, imaging characteristics and blood measurements of the 182 included patients are shown in Tables 1 and 2. Seventy-four percent of the patients were male and the mean age was $67 \pm 9$ years. Prior to the index event, statins were used by 51% of the patients, 43% of the patients used antiplatelet medication and 3% used anticoagulants. Presence of ulceration, IPH, LRNC and TRFC in the symptomatic carotid plaque was found in 28%, 39%, 64% and 38% of the patients, respectively. Of the patients with ulcerations, 23% had ulcerations of 1 mm, 39% had ulcerations between 1 and 2 mm and 39% had ulcerations > 2 mm. The median relative volumes of IPH and LRNC were 0.0 [0.0–4.7]% and 2.3 [0.0–11.3]%, respectively. Calcifications were present in 91% of the patients, with a median relative volume of 2.3 [0.4–6.5]%. Plaque volume had a median of 1217.9 [994.8–1469.3] mm$^3$. NETs levels (measured as MPO-DNA complexes) showed a median of 23 [4–96] mAU. Since NETs levels were not normally distributed, patients were classified as having low or high levels, with the cut-off at 50% of the observations. The general marker of cell-death, histone-DNA, showed median levels of 69 [46–100] mAU. These two levels were weakly, but significantly correlated (Spearman's correlation coefficient = 0.32, $p<0.001$).

### Lack of association between plaque vulnerability and plasma levels of NETs

First, the vulnerable plaque characteristics were submitted to a principal component analysis to construct one 'vulnerability index'. The principal component with the highest eigenvalue

**Table 1. Clinical characteristics.**

| Clinical characteristics | Total population (n = 182) |
|---|---|
| Age (years) | 67 ± 9 |
| Sex (male) | 135 (74%) |
| BMI | 27 ± 4 |
| Classification event | - |
| TIA | 77 (42%) |
| Stroke | 82 (45%) |
| Amaurosis fugax | 23 (13%) |
| Cause of the ischemic event | |
| Large-artery atherosclerosis | 161 (88%) |
| Stroke of undetermined etiology | 21 (12%) |
| Diabetes mellitus | 44 (24%) |
| Hypercholesterolemia | 142 (78%) |
| Hypertension | 129 (71%) |
| History of cardiovascular disease | 89 (49%) |
| Current smoking | 39 (21%) |
| Medication use prior to event | - |
| Statins | 92 (51%) |
| Antihypertensive drugs | 111 (61%) |
| Antidiabetic drugs | 33 (18%) |
| Antiplatelet drugs | 79 (43%) |
| Anticoagulants | 5 (3%) |

Data is presented as mean (SD) for normally distributed variables, median [25th-75th percentile] for not normally distributed variables and number (percentage) for frequencies. BMI, body mass index; TIA, transient ischemic attack. Cause of the ischemic event is according to the TOAST classification.

explained 48.1% of total variance in the data (see Table 3 and S2 Fig). This component comprised all known vulnerable plaque characteristics, and was negatively affected by relative calcification volume. Therefore, we defined this component as the vulnerability index of the atherosclerotic plaques. Furthermore, patients were classified as having low or high levels of NETs, and this classification was used as dependent variable in logistics regression models. The vulnerability index was positively, but not statistically significantly associated with high levels of NETs (odds ratio [OR] 1.28, 95% confidence interval [CI] 0.90–1.83).

## Association of plaque vulnerability with plasma levels of NETs in patients without statins or antithrombotic medication

Since we observed significant interactions between vulnerable plaque characteristics and use of medication prior to the index event, subgroup analyses were performed in patients with or without statins or antithrombotic medication use prior to the index event. Patient characteristics of both subgroups are shown in S1 Table. Patients naive to statins or antithrombotic medication prior to the index event (n = 72) were significantly younger, were more often female and had less co-morbidities compared to patients who were using statins or antithrombotic medication prior to the index event (n = 109). NETs levels (MPO-DNA) were not significantly different between the two subgroups. The principal components identified in both subgroups were similar to those identified in the total population (see S2 Table). In the subgroup of patients naive to statins or antithrombotic medication, the vulnerability index was positively

**Table 2. Imaging biomarkers and blood measurements.**

| Imaging biomarkers (symptomatic artery) | Total population (n = 182) |
|---|---|
| Degree of stenosis (ECST) (%) | 55 ± 16 |
| **MDCTA** (n = 160) | - |
| Interval event-MDCTA (days) | 32 [12–52] |
| Interval MDCTA-blood withdrawal (days) | 0 [0–24] |
| Presence plaque ulceration | 44 (28%) |
| Plaque ulceration size = 1 mm | 10 (23%) |
| Plaque ulceration size >1 and ≤2 mm | 17 (39%) |
| Plaque ulceration size >2 mm | 17 (39%) |
| Presence calcifications | 144 (91%) |
| Relative calcification volume (%) | 2.3 [0.4–6.5] |
| **MRI** (n = 172) | - |
| Interval event-MRI (days) | 47 [31–67] |
| Interval MRI-blood withdrawal (days) | 0 [−1–0] |
| Presence IPH | 67 (39%) |
| Relative IPH volume (%) | 0.0 [0.0–4.7] |
| Presence LRNC | 108 (64%) |
| Relative LRNC volume (%) | 2.3 [0.0–11.3] |
| Thin or ruptured fibrous cap | 64 (38%) |
| Plaque volume (mm$^3$) | 1217.9 [994.8–1469.3] |
| **Blood measurements** | - |
| Interval event-blood withdrawal (days) | 46 [29–67] |
| MPO-DNA (mAU) | 23 [4–96] |
| Histone-DNA (mAU) | 69 [46–100] |

Data is presented as mean (SD) for normally distributed variables, median [25th-75th percentile] for not normally distributed variables and number (percentage) for frequencies. AU, arbitrary units; ECST, European Carotid Surgery Trial; IPH, intraplaque hemorrhage; LRNC, lipid-rich necrotic core; MDCTA, multidetector-row computed tomography; MPO, myeloperoxidase; MRI, magnetic resonance imaging.

**Table 3. Varimax Rotated Component Matrix derived from PCA.**

| | Factor loadings in 'vulnerability index' component |
|---|---|
| Relative LRNC volume (%) | **0.944** |
| Relative IPH volume (%) | **0.905** |
| Thin or ruptured fibrous cap | **0.766** |
| Ulceration size | **0.535** |
| Relative calcification volume (%) | -0.203 |
| Plaque volume (mm$^3$) | **0.510** |
| *Eigenvalue* | 2.884 |
| *Variance explained (%)* | 48.1 |

The different vulnerable plaque components were combined into components using principal component analysis, using the varimax rotation method. The component with the highest eigenvalue was selected, which represents all important vulnerable plaque characteristics. This component was used in subsequent analyses to investigate the association between plaque vulnerability and MPO-DNA or histone-DNA levels.

Abbreviations: IPH, intraplaque hemorrhage; LRNC, lipid-rich necrotic core; PCA, principal component analysis. Bold values represent highest factor loadings.

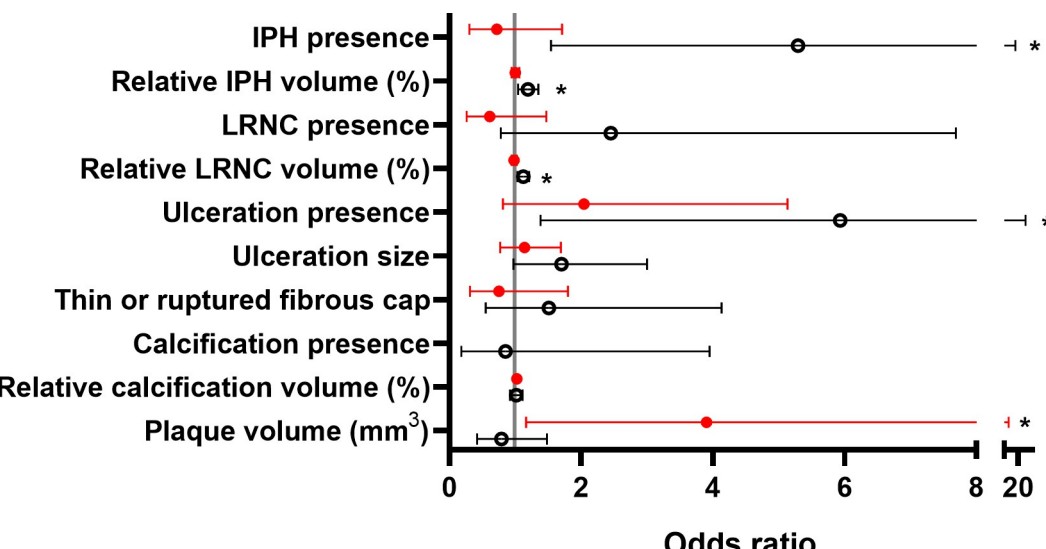

**Fig 1. Association between plaque characteristics and MPO-DNA levels in subgroups stratified by medication use prior to the index event.** Logistic regression with two categories of MPO-DNA as dependent variable (high vs low) and plaque characteristics as independent variables, adjusted for age, sex and time between index event and blood sampling. Odds ratios with 95% confidence intervals are reported for the two subgroups. Odds ratio for plaque volumes is presented for 1000 mm³. *indicates *p*-value below 0.05. IPH, intraplaque hemorrhage; LRNC, lipid-rich necrotic core.

associated with high levels of NETs (OR 2.08, 95% CI 1.04–4.17). In the subgroup of patients who did use statins or antithrombotic medication prior to the index event, no significant association was found between NETs levels and the vulnerability index (OR 1.10, 95% CI 0.68–1.79).

Finally, to investigate which individual plaque characteristics contribute to this association between vulnerability and NETs levels in the group of patients naive to statins or antithrombotic medication, the associations between individual plaque characteristics and plasma levels of NETs were investigated (Fig 1 and S3 Table). Multiple significant associations were found between vulnerable plaque characteristics and NETs levels in the group of patients naive to statins or antithrombotic medication. The presence of IPH (OR 5.29, 95% CI 1.54–18.06), relative volume of IPH (OR 1.19, 95% CI 1.04–1.35), relative volume of LRNC (OR 1.12, 95% CI 1.03–1.21) and presence of ulceration (OR 5.93, 95% CI 1.38–25.37) were significantly associated with high NETs levels (Fig 1 and S3 Table). In the group of patients who did use statins or antithrombotic medication prior to the index event, no significant associations between vulnerable plaque characteristics and NETs levels were found, except for a positive significant association between plaque volume (per 1000 mm³) and NETs levels (OR 3.90, 95% CI 1.16–13.13).

## Lack of association between plaque vulnerability and histone-DNA levels

Finally, the association between plaque vulnerability and histone-DNA levels (marker for more general cell death) was investigated. No significant associations between the vulnerability index and histone-DNA levels were found, neither in the total population (OR 0.86, 95% CI 0.60–1.24) nor in the subgroups with (OR 0.97, 95% CI 0.59–1.58) or without statin or antithrombotic medication use prior to the index event (OR 2.75, 95% CI 0.77–9.86).

## Discussion

The main finding of this study in patients with symptomatic carotid atherosclerotic plaques is the positive association between the vulnerability index and NETs levels in the subgroup of patients naive to statins and antithrombotic medication prior to the index event. Subsequent analyses showed that this association is mainly driven by the presence and volume of IPH, volume of LRNC and the presence of ulceration. To our knowledge, this is the first study that investigated the association between plaque vulnerability and NETs levels in patients with a previous ischemic event.

The first component generated by our principal component analysis included all measured vulnerable plaque characteristics: IPH, LRNC, ulceration, TRFC and plaque volume. Calcification was found to negatively affect this component, which conforms to previous literature describing an inverse association between calcification and plaque vulnerability [26]. In patients with suspected coronary artery disease, a positive association between MPO-DNA complexes was reported with both the number of atherosclerotic coronary vessels and the occurrence of major adverse cardiac events, suggesting a relationship between NETs levels and unstable plaques [22]. However, in our patient population, no significant association between the vulnerability index and NETs levels was found, except in patients naive to statin or antithrombotic medications prior to the index event.

Indeed, when we stratified patients by the use of statins or antithrombotic medication prior to the index event, we found a positive association between the vulnerability index and NETs levels in patients naive to this medication, while no significant association was observed in the subgroup of patients who did use this medication prior to the index event. This can possibly be explained by the pleiotropic effects of statins and antithrombotic medication on atherosclerotic plaque vulnerability, inflammation, endothelial function and thrombosis. Indeed, statins and aspirin, the most commonly used lipid lowering and antiplatelet medication, are suggested to increase the stability of atherosclerotic plaques through their effects on inflammation, thrombosis and endothelial function [27–31]. On the other hand, antiplatelet medication is also associated with the presence of IPH, thereby potentially destabilizing plaques [32]. Furthermore, there are indications that both statins and antithrombotic medication affect levels of NETs, due to their anti-inflammatory effects [33–35]. Statins are shown to inhibit neutrophil migration to sites of inflammation, endothelial adhesion and neutrophil activation as a result of decreased cytokine and chemokine production [36–38]. The antiplatelet drug acetylsalicylic acid decreased NET formation *in vitro* and in mice, potentially via inhibiting the activation of nuclear factor-κB [39]. This could explain why we could only detect the association between plaque vulnerability and NETs in patients who did not use this medication. Long-term medication-use in other patients probably attenuated this association.

The patients using medication prior to the index event were significantly older, were more often male and showed an increased prevalence of co-morbidities, which could also have influenced the association between NETs and vulnerable plaque characteristics (see S1 Table). However, the level of NETs (MPO-DNA) was not significantly different between the subgroups. It should be noted that most patients naive to statins and antithrombotic medication prior to the index event started using these medications after the index event, but before blood sampling and imaging in this study. While this could have affected plaque characteristics and NETs levels, it most likely would have resulted in underestimation of our results at the time of blood sampling. Overall however, this short-term use of statins or antithrombotic medication is not expected to result in large changes in plaque characteristics [40].

Subsequent analyses investigating the association between individual vulnerable plaque characteristics and NETs levels revealed a positive association between ulceration and high

NETs levels in patients naive to statins and antithrombotic medication prior to the index event. Ulceration, the presence of cavities in the surface of atherosclerotic plaques, is known to be associated with increased overall vulnerability and rupture of plaques, and with the risk of ischemic events [20, 41]. It has been suggested that ulcerations reflect sites of previous rupture of the atherosclerotic plaque [42]. Our data can not reveal whether high levels of NETs are the cause or consequence of ulceration. Ulceration possibly leads to increased inflammation and release of NETs. Also, inflammation leading to neutrophil activation and high NETs levels can result in more vulnerable plaques, increasing the chance of ulceration [6, 43].

The presence and volume of IPH also showed a positive association with NETs levels in patients naive to statins and antithrombotic medication. IPH can result from blood leaking from immature neovessels formed inside the atherosclerotic plaque as a response to hypoxia, or from vessels in regression or damaged vessels [44]. Increased inflammation can therefore aggravate IPH as a consequence of the high oxygen consumption of inflammatory cells, which induces more angiogenesis and therefore potentially more bleeding [45]. However, recent research found no positive association between plaque microvasculature and IPH [46]. In addition, results from previous research suggest that previous plaque rupture can contribute to the development of IPH [18, 47]. The increased influx of (activated) inflammatory cells inside the plaque, which is associated with IPH, might result in increased release of NETs [48]. Recently, in plaques obtained by autopsy on myocardial infarction patients, higher levels of NETs have been shown to be present in plaques with IPH compared to intact plaques, which is in accordance with our results measured in plasma [49]. It should be noted that the confidence intervals for our found associations are quite wide, which makes it difficult to reach clear conclusions about the size of the effects found. This is probably caused by the relatively small sample size, therefore further research in a larger cohort is needed.

In this study, we measured the levels of MPO-DNA complexes and histone-complexed DNA fragments using ELISA assays. We used the MPO-DNA levels as a measure for the NETs levels. Currently, there is no gold standard for detection of NETs, although ELISA assays measuring NET by-products, most commonly MPO-DNA complexes, are currently believed to be the most objective and specific detection method [50, 51]. We observed relatively low levels of MPO-DNA complexes in a large number of patients. Recently, we showed that MPO-DNA complexes measured in healthy subjects without symptomatic atherosclerosis were generally very low [24]. Previously, histone-complexed DNA fragments have also been used as markers for NETs. However, they are released upon cell death in general rather than being highly selective markers for NETs [52]. Extensive cell death may for example occur during inflammation or other conditions in which cellular damage takes place. Regardless, we did include the histone-DNA levels in our study. However, where we found significant associations between vulnerable plaque characteristics and MPO-DNA, no associations were observed between vulnerable plaque characteristics and histone-DNA. This again shows that there is no clear correlation between both markers.

Our study has some limitations. First, we had a relatively small sample size, which reduces power and limits the subgroup analyses and number of possible co-variates. Another limitation is that while we excluded all patients with cardioembolic stroke, we cannot be completely certain that the index event in all patients was caused by carotid artery disease. However, in the majority of patients (88%), large-artery atherosclerosis was the probable or definite cause of the ischemic event. Still, if the group with undetermined etiology had other causes than large-artery atherosclerosis, this might have affected our results. Therefore, replication of our results is needed, preferably in a larger population, while knowing for sure that the ischemic event is caused by the carotid artery disease.

## Future perspectives and clinical relevance

It should be noted that the median time interval between the index event, and blood sampling and imaging of the plaque is 46 days. Blood sampling and imaging of the plaque were performed at the same time. Therefore, while plasma levels of NETs do not represent the levels during the acute phase of the index event, they do reflect the characteristics of the plaque at the time of imaging. Our results suggest that the levels of NETs measured after the acute phase of an ischemic event might be predictive for the vulnerability of plaques, and therefore potentially for the risk of recurrent ischemic events. It is known that patients with a recent TIA or ischemic stroke are at a high risk for recurrent ischemic events. Currently, the degree of stenosis of the carotid artery in patients is the main determinant used to identify patients who will benefit from carotid endarterectomy [53]. To improve clinical decision-making regarding the need for carotid endarterectomy, it has been suggested that vulnerable plaque characteristics might help predict which patients with mild-to-moderate carotid artery stenosis are at highest risk for recurrent events [14]. While in the current study we observed associations between vulnerable plaque characteristics and NETs in patients naive to statins and antithrombotic medication prior to the index event, further research should address whether NETs levels measured after the acute phase of an ischemic event could predict recurrent ischemic events and could therefore be useful in the decision-making process.

## Conclusion

In conclusion, the vulnerability of atherosclerotic plaques is positively associated with NETs levels in patients with a symptomatic mild-to-moderate carotid artery stenosis, but only in the subgroup of patients naive to statins and antithrombotic medication prior to the index event. This suggests long-term usage of this medication affects plaque stability and release of NETs in such a way that their association is attenuated. These findings highlight the association between atherosclerosis and NETs and further elucidate mechanisms underlying plaque vulnerability.

## Supporting information

**S1 Fig. Distribution of MPO-DNA complex and histone-DNA levels and classification into two groups.** (A) Histograms of MPO-DNA complex and histone-DNA levels. (B) Median levels with interquartile range of MPO-DNA complex and histone-DNA for groups with low and high levels. To be able to use MPO-DNA or histone-DNA levels as dependent variable in the logistic regression models, the cut-off value was set at 50% of the values, resulting in a cut-off of 23 mAU for MPO-DNA and 69 mAU for histone-DNA. mAU, milli-arbitrary units; MPO, myeloperoxidase.
(DOCX)

**S2 Fig. Scree plot for the principal component analysis as presented in Table 3.** The different vulnerable plaque components were combined into components, using the varimax rotation method. Component 1 has the highest eigenvalue and was therefore used as 'vulnerability index' in this study.
(DOCX)

**S1 Table. Clinical characteristics in the subgroups of patients stratified by statin and antithrombotic medication use prior to the index event.** Data is presented as mean (SD) for normally distributed variables, median [25th-75th percentile] and number (percentage) for frequencies. P-value represents the difference between patients with and without use of statins or antithrombotic medication tested by the independent students' t-test, Mann-Whitney U

test or Chi-square test. *indicates p-value below 0.05. Abbreviations: BMI, body mass index; mAU, milli-arbitrary units; MPO, myeloperoxidase; NETs, neutrophil extracellular traps; TIA, transient ischemic attack.
(DOCX)

**S2 Table. Varimax Rotated Component Matrix derived from PCA for the subgroups of patients stratified by statin and antithrombotic medication use prior to the index event.** The different vulnerable plaque characteristics were combined into one component representing the vulnerability of plaques using principal component analysis, while stratified in two groups based on the use of statin and antithrombotic medication prior to the index event. Similar components were generated in both groups, which were also comparable to the component generated in the total population (Table 3). Abbreviations: IPH, intraplaque hemorrhage; LRNC, lipid-rich necrotic core; PCA, principal component analysis. Bold values represent highest factor loadings per component.
(DOCX)

**S3 Table. Association between plaque characteristics and MPO-DNA levels in subgroups stratified by medication use prior to the index event.** Logistic regression with two categories of MPO-DNA as dependent variable (high vs low) and plaque characteristics as independent variables, adjusted for age, sex and time between index event and blood sampling. OR for plaque volumes is presented for 1000 mm3. *indicates p-value below 0.05. CI, confidence interval; IPH, intraplaque hemorrhage; LRNC, lipid-rich necrotic core; OR, odds ratio.
(DOCX)

## Acknowledgments

We thank Debby Priem-Visser for her excellent technical assistance.

## Author Contributions

**Conceptualization:** Judith J. de Vries, Anouchska S. A. Autar, Dianne H. K. van Dam-Nolen, Samantha J. Donkel, Daniel Bos, Aad van der Lugt, Moniek P. M. de Maat, Heleen M. M. van Beusekom.

**Formal analysis:** Judith J. de Vries, Anouchska S. A. Autar, Dianne H. K. van Dam-Nolen, Samantha J. Donkel, Daniel Bos, Aad van der Lugt, Moniek P. M. de Maat, Heleen M. M. van Beusekom.

**Investigation:** Dianne H. K. van Dam-Nolen, Mohamed Kassem, Anja G. van der Kolk, Twan J. van Velzen, M. Eline Kooi, Jeroen Hendrikse, Paul J. Nederkoorn, Aad van der Lugt.

**Resources:** Dianne H. K. van Dam-Nolen, Mohamed Kassem, Anja G. van der Kolk, Twan J. van Velzen, M. Eline Kooi, Jeroen Hendrikse, Paul J. Nederkoorn, Aad van der Lugt.

**Supervision:** Aad van der Lugt, Moniek P. M. de Maat, Heleen M. M. van Beusekom.

**Visualization:** Judith J. de Vries.

**Writing – original draft:** Judith J. de Vries, Anouchska S. A. Autar.

**Writing – review & editing:** Judith J. de Vries, Anouchska S. A. Autar, Dianne H. K. van Dam-Nolen, Samantha J. Donkel, Mohamed Kassem, Anja G. van der Kolk, Twan J. van Velzen, M. Eline Kooi, Jeroen Hendrikse, Paul J. Nederkoorn, Daniel Bos, Aad van der Lugt, Moniek P. M. de Maat, Heleen M. M. van Beusekom.

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
