## [Editor Report · Decision Letter 0]

29 Dec 2021

PONE-D-21-22134Association between plaque vulnerability and neutrophil extracellular traps (NETs) levels: the Plaque At RISK studyPLOS ONE

Dear Dr. de Vries,

Thank you for submitting your manuscript to PLOS ONE. After careful consideration, we feel that it has merit but does not fully meet PLOS ONE’s publication criteria as it currently stands. Therefore, we invite you to submit a revised version of the manuscript that addresses the points raised during the review process. Find attached some minor comments in terms of methodology and results section, that should be considered.

We look forward to receiving your revised manuscript.

Kind regards,

Miguel A. Barboza, MD, MSc

Academic Editor

PLOS ONE

Journal Requirements:

[I have read the journal's policy and the authors of this manuscript have the following competing interests: MEK reports grants outside the submitted work from NWO Aspasia, NWO Hestia, and EU Horizon 2020 ITN and has research collaborations with PieMedical Systems, Machnet BV.] 

Reviewer Comments:

I had the pleasure to review the manuscript “Association between plaque vulnerability and neutrophil extracellular 2 traps (NETs) levels: the Plaque At RISK study”, which investigate variables associated with plaque rupture and NETs potential relationship. The topic is novel, and in general the manuscript is well written and the methodology is strong and well conducted to explain the association.

Before considering this manuscript for publication, I have some minor comments to add:

1. Abstract: please add p values from the associations mentioned, in line 50 and 52

2. Introduction: nothing to add

3. Methods:

a. Despite it is described in the original protocol, I suggest the authors to clarify in this paper the definition of “minor stroke” (NIHSS?? Size of lesion?? Etc), to facilitate the reader’s experience.

b. When including information from TIA or stroke, did the authors from the original study or the current manuscript, performed investigation methodology in terms of explanation of the index event and possible association of the atherosclerotic vessel (such as TOAST, ASCOD, etc); in other words, are you sure that all the index cerebrovascular events were related to the carotid disease? (for example, if there was a lacunar stroke, hypertension as risk factor could have a higher association than atherosclerosis); I suggest to clarify cerebrovascular events characteristics (median NIHSS, ABCD2 score for TIA, and ASCOD/TOAST classification for those cases included).

4. Results:

a. Why 3% of cases were under oral anticoagulation? If so, and they had a new TIA/stroke, how certain you are that the carotid disease could be related as the risk factor for this event (low INR for those taking VKA or sub-therapeutic dosage for DOAC?)

5. Discussion

a. A few words in terms of limitations in the presence study should be acknowledged, mainly in sample size, co-variate analysis and type of vascular lesions (knowing that the index stroke/TIA is caused by the stenotic vessel as the main etiology).

---

## [Author Response · Author response to Decision Letter 0]

9 Feb 2022

Journal Requirements:

Answer: We changed the file names of the supporting files and made some adjustments in fonts following the style templates. 

[I have read the journal's policy and the authors of this manuscript have the following competing interests: MEK reports grants outside the submitted work from NWO Aspasia, NWO Hestia, and EU Horizon 2020 ITN and has research collaborations with PieMedical Systems, Machnet BV.] 

Answer: We included our updated Competing Interests statement in our cover letter.

Answer: We updated our Data Availability statement in the manuscript and included the updated information in our cover letter:

‘The data underlying this article cannot be shared publicly due to the privacy of individuals that participated in this study. The data will be shared on reasonable request to the Radiology trial office (imaging.trialbureau@erasmusmc.nl)..’

Answer: We completed reference 21. 

We added reference 28, which is a correction to reference 27. We decided to keep reference 27 in the manuscript, since the correction only involves a conflicts of interest statement.

Reviewer Comments:

I had the pleasure to review the manuscript “Association between plaque vulnerability and neutrophil extracellular 2 traps (NETs) levels: the Plaque At RISK study”, which investigate variables associated with plaque rupture and NETs potential relationship. The topic is novel, and in general the manuscript is well written and the methodology is strong and well conducted to explain the association.

Before considering this manuscript for publication, I have some minor comments to add:

1. Abstract: please add p values from the associations mentioned, in line 50 and 52

Answer: Thank you for your willingness to review our manuscript. We added the p-values from the mentioned associations in line 50 and 52.

2. Introduction: nothing to add

3. Methods:

a. Despite it is described in the original protocol, I suggest the authors to clarify in this paper the definition of “minor stroke” (NIHSS?? Size of lesion?? Etc), to facilitate the reader’s experience.

Answer: We added the definition of ‘minor stroke’ in line 100 as follows: “(modified Rankin scale ≤3)”.

b. When including information from TIA or stroke, did the authors from the original study or the current manuscript, performed investigation methodology in terms of explanation of the index event and possible association of the atherosclerotic vessel (such as TOAST, ASCOD, etc); in other words, are you sure that all the index cerebrovascular events were related to the carotid disease? (for example, if there was a lacunar stroke, hypertension as risk factor could have a higher association than atherosclerosis); I suggest to clarify cerebrovascular events characteristics (median NIHSS, ABCD2 score for TIA, and ASCOD/TOAST classification for those cases included).

Answer: We thank the reviewer for this comment. The patients included in this study were part of the PARISK study (Plaque At RISK; clinical trials.gov NCT01208025). Patients who had a stroke probably caused by cardiac embolism or clotting disorders were excluded from the PARISK study. From 88% of the included patients, large-artery atherosclerosis (LAA) was the probable or definite cause of the cerebrovascular event, according to the TOAST classification. For the other 21 patients, the cause was unknown or a combination of probable causes (excluding cardiac embolism and clotting disorders) was reported. We therefore assume that we have a population of patients in whom the index cerebrovascular event is highly related to the carotid disease. We included an overview of the probable causes of the index event in Table 1 to provide some more information on this. Also we did include this as a limitation in our discussion (line 385-393, see below).

We agree with the reviewer that the NIHSS would indeed also have been valuable. However, the patients were included in the PARISK study almost 10 years ago, when we did not score NIHSS for every patient yet.

The modified Rankin Score was available for all patients before the event and at discharge. However, this score only represents the degree of disability of patients and does not say something about the event itself, hence we do not think it will add to the manuscript and will therefore not include it in the manuscript. However, for your information, we show the distribution of the mRS in our population below:

mRS score Before the event After discharge

0 (no symptoms) 141 (79%) 79 (44%)

1 (minor symptoms) 20 (11%) 62 (35%)

2 (minor handicaps) 16 (9%) 29 (16%)

3 (moderate handicap) 2 (1%) 8 (5%)

4. Results:

a. Why 3% of cases were under oral anticoagulation? If so, and they had a new TIA/stroke, how certain you are that the carotid disease could be related as the risk factor for this event (low INR for those taking VKA or sub-therapeutic dosage for DOAC?)

Answer: Indeed 3% of the cases (5 patients) were under oral anticoagulation, mainly because of a history of cardiovascular disease. It should be noted that it is still possible to develop a cerebrovascular event while using oral anticoagulation. 

In addition, we checked the categories of causes for the ischemic events in these 5 patients: 3 patients showed indications for large-artery atherosclerosis as probable or definite cause, and 2 patients were classified as having an unknown cause of the ischemic event. 

We agree that the use of oral anticoagulation is a relevant characteristic, to the same degree as other medications used. Therefore, we checked the interaction of medication with our results. Indeed, in our subgroup analysis, we observed significant associations between levels of MPO-DNA and the vulnerability of the plaque in patients who did not use statins or antithrombotic medication prior to the index event. This association was not observed in patients who did use these types of medication before the index event. This shows that medication use, among which was oral anticoagulation, does affect the association between MPO-DNA levels (NETs) and plaque vulnerability. 

5. Discussion

a. A few words in terms of limitations in the presence study should be acknowledged, mainly in sample size, co-variate analysis and type of vascular lesions (knowing that the index stroke/TIA is caused by the stenotic vessel as the main etiology).

Answer: We added a few words about limitations of our study in lines 385-393: 

“Our study has some limitations. First, we had a relatively small sample size, which reduces power and limits the subgroup analyses and number of possible co-variates. Another limitation is that while we excluded all patients with cardioembolic stroke, we cannot be completely certain that the index event in all patients was caused by carotid artery disease. However, in the majority of patients (88%), large-artery atherosclerosis was the probable or definite cause of the ischemic event. Still, if the group with undetermined etiology had other causes than large-artery atherosclerosis, this might have affected our results. Therefore, replication of our results is needed, preferably in a larger population, while knowing for sure that the ischemic event is caused by the carotid artery disease.’

---

## [Editor Report · Decision Letter 1]

30 May 2022

Association between plaque vulnerability and neutrophil extracellular traps (NETs) levels: the Plaque At RISK study

PONE-D-21-22134R1

Dear Dr. de Vries,

We’re pleased to inform you that your manuscript has been judged scientifically suitable for publication and will be formally accepted for publication once it meets all outstanding technical requirements.

Kind regards,

Giuseppe Danilo Norata

Academic Editor

PLOS ONE

Additional Editor Comments (optional):

All comments have been properly addressed
---

## [Editor Report · Acceptance letter]

1 Jun 2022

PONE-D-21-22134R1 

Association between plaque vulnerability and neutrophil extracellular traps (NETs) levels: the Plaque At RISK study 

Dear Dr. de Vries:

I'm pleased to inform you that your manuscript has been deemed suitable for publication in PLOS ONE. Congratulations! Your manuscript is now with our production department. 

Kind regards, 

on behalf of

Dr. Giuseppe Danilo Norata 

Academic Editor

PLOS ONE